

# A multi-method approach for assessing the distribution of a rare, burrowing North American crayfish species

Kathleen B. Quebedeaux[1,2], Christopher A. Taylor[1], Amanda N. Curtis[3] and Eric R. Larson[2]

[1] Illinois Natural History Survey, Prairie Research Institute, University of Illinois at Urbana-Champaign, Champaign, IL, United States of America

[2] Department of Natural Resources and Environmental Sciences, College of Agricultural, Consumer and Environmental Sciences, University of Illinois at Urbana-Champaign, Urbana, IL, United States of America

[3] Program in Ecology, Evolution & Conservation Biology, University of Illinois at Urbana-Champaign, Urbana, IL, United States of America

Corresponding author
Kathleen B. Quebedeaux, kbquebedeaux@gmail.com

## ABSTRACT

Primary burrowing crayfishes face high extinction risk, but are challenging to study, manage, and conserve due to their difficult-to-sample habitat (*i.e.*, terrestrial burrows) and low population densities. We apply here a variety of methods to characterize the distribution, habitat associations, and conservation status of the Boston Mountains Crayfish *Cambarus causeyi* (Reimer, 1966), an endemic burrowing crayfish found only in the Ozark Mountains of Arkansas, United States. We used species distribution modeling (SDM) on historic occurrence records to characterize the distribution and macroscale habitat associations of this species. We then ground-truthed SDM predictions with conventional sampling, modeled fine-scale habitat associations with generalized linear models (GLM), and lastly developed and tested an environmental DNA (eDNA) assay for this species in comparison to conventional sampling. This represents, to our knowledge, the first successful eDNA assay for a terrestrial burrowing crayfish. Our MaxEnt-derived SDM found a strong effect of average annual precipitation on the historic distribution of *C. causeyi*, which occurred most frequently at locations with moderately high average annual precipitation (140–150 cm/yr) within our study region. *Cambarus causeyi* was difficult to detect by conventional sampling in 2019 and 2020, found at only 9 of 51 sites (17.6%) sampled by searching for and manually excavating crayfish burrows. Surprisingly, habitat suitability predicted from our MaxEnt models was not associated with contemporary *C. causeyi* occurrences per GLMs. Instead, *C. causeyi* presence was negatively associated with both sandy soils and the presence of other burrowing crayfish species. Poor SDM performance in this instance was likely caused by the omission of high resolution fine-scale habitat data (*e.g.*, soils) and biotic interactions from MaxEnt models. Finally, our eDNA assay detected *C. causeyi* from six of 25 sites (24.0%) sampled in 2020, out-performing conventional surveys by burrow excavation for this species. Given the difficulty of studying primary burrowing crayfishes and their high conservation need, we propose that eDNA may become an increasingly important monitoring tool for *C. causeyi* and similar species.

# INTRODUCTION

Primary burrowing crayfishes are crayfish that spend the majority of their lives in burrows and only leave occasionally to mate and find food (*Hobbs Jr, 1942*). Burrowing crayfish play an important role as ecosystem engineers by disturbing the soil in which they burrow and by creating habitat for a wide range of other taxa (*Pintor & Soluk, 2006*; *Welch et al., 2008*). Crayfish are understudied relative to other aquatic taxa (*Reid et al., 2019*), and even among crayfishes, primary burrowing crayfish are particularly understudied (*Bloomer, DiStefano & Taylor, 2021*; *Moore, DiStefano & Larson, 2013*). About one third of all crayfish species globally are threatened with extinction (*Richman et al., 2015*), and about 22% of North American crayfish species of conservation concern are primary burrowing species (*Taylor et al., 2007*). The high rate of endemism and narrow ranges among crayfish species exacerbates their imperilment rate (*Dudgeon et al., 2006*; *Taylor et al., 1996*). Additionally, primary burrowing crayfish are known to inhabit a wide diversity of habitats such as open-canopied grasslands (*Rhoden, Taylor & Peterman, 2016*), pitcher plant bogs (*Welch et al., 2008*), and upland forested seeps (*Loughman, 2010*), and can be habitat specialists or generalists (*Loughman, Welsh & Simon, 2012*). To effectively conserve a species, resource managers must understand its distribution and habitat requirements in order to make informed decisions (*Richter et al., 1997*; *Taylor et al., 2007*).

Species distribution modeling (SDM) is one tool for analyzing the habitat variables driving the range of rare and endemic taxa (*Fois et al., 2018*; *Peterman, Crawford & Kuhns, 2013*; *Rhoden, Peterman & Taylor, 2017*). Species distribution models use species locality and environmental data to make correlative models depicting relative habitat suitability for focal organisms (*Warren & Seifert, 2011*). A common and effective approach for creating SDMs is maximum entropy modeling or MaxEnt (*Elith et al., 2011*; *Phillips, Anderson & Schapire, 2006*). MaxEnt is useful for rare and understudied species because it performs well with small sample sizes and is capable of using presence-only data (*Elith et al., 2011*; *Hernandez et al., 2006*; *Rhoden, Peterman & Taylor, 2017*).

While SDMs can be used for analyzing the drivers of a species' distribution, ground-truthing or validation should be included as an important part of the process (*Egly & Larson, 2018*; *Peterman, Crawford & Kuhns, 2013*; *Rhoden, Peterman & Taylor, 2017*; *Stirling et al., 2016*). Habitat modeling using traditional field sampling can also incorporate variables influencing a species' distribution that are not easily accounted for in large-scale spatial data (*Hirzel et al., 2006*). This is particularly true for sessile organisms or organisms with a small home range, which benefit from finer grain analyses (*Guisan et al., 2007*). Species distribution models can poorly account for the influence of biotic interactions, lack fine-scale habitat information, or be limited by biased or unreliable historical data (*Fourcade et al., 2014*; *Guisan & Thuiller, 2005*; *Peterman, Crawford & Kuhns, 2013*). These potential

limitations of SDMs illustrate the need for incorporating ground-truthing and utilizing traditional habitat sampling when studying a species' habitat needs and distribution.

Another emerging method of detecting rare, endemic taxa is employing molecular approaches collectively termed "environmental DNA" (eDNA). Environmental DNA refers to the capture and identification of DNA from an environmental sample (air, soil, or water), generally without the collection or identification of entire organisms (*Rees et al., 2014*). Environmental DNA may detect some cryptic or low-abundance organisms better than conventional sampling methods (*Barnes & Turner, 2016*). As a recent example, *Rice, Larson & Taylor (2018)* found eDNA to successfully detect a rare, large-river crayfish in the Missouri Ozarks. However, eDNA has not yet been used to detect primary burrowing crayfish, which are often more difficult to collect than their stream dwelling counterparts (*Taylor, Engelbert & DiStefano, 2015*). Environmental DNA offers a unique opportunity to detect primary burrowing crayfish, while avoiding the high amounts of effort and low detection rates associated with traditional methods, such as hand excavating burrows.

The Boston Mountains Crayfish (*Cambarus causeyi Reimer, 1966*) is an understudied and rarely observed burrowing crayfish endemic to the Boston Mountains of Arkansas, United States of America (USA). *Cambarus causeyi* spends the majority of its life in generally shallow, yet complex, burrows that are often found underneath large rocks (*Robison, Crandall & Wagner, 2009*; *Robison & Leeds, 1996*). Occasionally *C. causeyi* occurs in "small springs and tiny creeks under rocks" when reproducing, and was noted to be associated with intermittent mountain seeps and streams (*Robison, Crandall & Wagner, 2009*; *Robison & Leeds, 1996*). Between 1941 and 1986, only nine *C. causeyi* individuals were captured over six surveys. These specimens were collected from four localities in Pope County and one locality in Stone County (*Robison & Leeds, 1996*). Subsequently, *Robison & Leeds (1996)* reported the capture of 87 individuals from 39 separate localities (erroneously reported as 40). *Robison & Leeds (1996)* expanded the known range of *C. causeyi* to include Madison, Johnson, Franklin, Newton, and Searcy counties. However, they reported that 67 of the individuals were collected from Johnson County. *Robison, Crandall & Wagner (2009)* conducted a follow-up study that located *C. causeyi* at four new localities, but otherwise could not recover the species from the vast majority of sites sampled by *Robison & Leeds (1996)*. The extreme decrease in the number of individuals caught in the 2009 study caused *C. causeyi* to be updated to the status of Vulnerable on the 2009 IUCN assessment (*Robison, Crandall & Wagner, 2009*). Only two additional records of *C. causeyi* have been made by Arkansas Game and Fish Commission (AGFC) since 2009; they are from Newton and Van Buren counties in 2017 and 2019, respectively (B Wagner, pers. comm., 2019).

We applied SDMs, fine-scale modeling, and eDNA to improve understanding of the distribution and habitat associations for *C. causeyi*. Combined, these methods were intended to help us determine if *C. causeyi* declined across its range as suggested in *Robison, Crandall & Wagner (2009)* and identify habitat needed to protect the species. We also evaluated whether SDM approaches like MaxEnt are an effective means of predicting occurrences of rare, difficult to detect burrowing crayfish. Additionally, we tested the efficacy of utilizing eDNA as a surveillance tool for *C. causeyi* and other primary burrowing

crayfish species. The results of this study will inform managers looking to monitor *C. causeyi* and primary burrowing crayfish as a whole.

## MATERIALS & METHODS

### Species distribution modeling

Using presence-only historical occurrences, we created a SDM of suitable habitat for *C. causeyi*, using MaxEnt (*v.* 3.4.1) with the methods previously described in *Quebedeaux (2021)*. Due to the vagueness of many of the historical site descriptions from *Robison & Leeds (1996)* and *Robison, Crandall & Wagner (2009)*, we sorted the 44 historical sites by the precision of their location descriptions. Out of the 44 sites there were two exact locations, eight good locations, 27 fair locations, two poor locations, and five unusable locations (Fig. S1). Exact locations were sites where geographic coordinates were provided by the original source. Good locations lacked coordinates but could be narrowed down to a specific location and no other locations matched the historical description. An example of a site that was classified as good was the "roadside seepage on St. Hwy. 16, 3.1 mi. S of jct. of St. Hwys. 16 and 23 and 0.4 mi. W. of Dutton" (*Robison, Crandall & Wagner, 2009*). Fair locations provided a township, range, and section, but an exact location was not identifiable. Fair locations were either placed on a road crossing at a stream that seemed to match the description, or in the center of the section if a specific stream crossing could not be selected. Poor locations did not provide a township, range, and section and had vague site descriptions. Unusable sites either had a description that made it unfeasible to locate the site, or were in the same township, range, and section as another vaguely identified site. Only sites labelled as exact, good, or fair were used in the model; this included 37 out of the 44 historical sites in total. Although we had a small number of samples in our SDM, MaxEnt has been shown to produce reliable models, even with smaller sample sizes (*Fois et al., 2018*; *Galante et al., 2018*).

Environmental variables included as predictors of *C. causeyi* habitat suitability in the SDM were precipitation, Euclidean distance from a stream, elevation, slope, solar radiation, and average available water storage for the top 150 cm of soil. Average annual precipitation data was taken from the 30-year Parameter-elevation Regressions on Independent Slopes Model (PRISM) data for 1981–2010 (*Daly, Taylor & Gibson, 1997*), and it was included in the model because of *C. causeyi*'s association with ephemeral, precipitation dependent streams (*Robison, Crandall & Wagner, 2009*). The Euclidean distance from a stream was calculated from the National Hydrography Dataset (NHD: *US Geological Survey, 2019*) and was selected because *C. causeyi* is associated with small streams. Elevation data were taken from the National Elevation Dataset (*US Geological Survey & EROS Data Center, 2019*), and we predicted that *C. causeyi* may be associated with seepages in high elevation areas. We predicted a connection between *C. causeyi* and higher elevation because stream discharge and velocity vary with elevation. Elevation has been shown to be a driver of other crayfish species' distributions, and *C. causeyi* has been noted to be found in higher elevation areas (*Dyer et al., 2013*; *Mouser, Mollenhauer & Brewer, 2019*; *Robison & Leeds, 1996*). Slope and solar radiation were calculated from the elevation dataset using ArcMap (*v.* 10.8.2) with

the ESRI Spatial Analysts Tools. Slope was included because we anecdotally noted that *C. causeyi* seemed to be found on steeper inclines. Solar radiation was used because it was found to be positively associated with other burrowing crayfish in Arkansas (*Rhoden, Peterman & Taylor, 2017*). Soil data were taken from the gridded Soil Survey Geographic Database (gSSURGO). We included average available water storage for the top 150 cm of soil because burrowing crayfish are associated with hydric environments (*Rhoden, Taylor & Peterman, 2016*). All environmental layers had a 30 x 30 m grain size. We identified an extent to model *C. causeyi*'s distribution by MaxEnt by selecting all Hydrologic Unit Codes 8 watersheds (*Watershed Boundary Dataset for HUC8, 2019*) that substantially overlapped with counties of historical records, clipping them to the boundaries of Arkansas, and omitting only one watershed with minimal overlap (Fig. S1).

We tested for correlations between all predictor variables using the Band Collection Statistics tool in ArcMap 10.8.2, and all variable combinations had a correlation coefficient less than 0.5. We next modeled suitable habitat for *C. causeyi* using a 45-replicate run in MaxEnt with bootstrapping for replication and a 20% random test percentage. Variable importance was measured with a jackknife analysis. We set a maximum of 5,000 iterations of the MaxEnt models with background predictions. The above settings and the defaults were used in the final MaxEnt model. Due to the narrow range of *C. causeyi* and the small number of presences, we did not use a bias file. We calculated percent contribution and permutation importance in order to assess variable importance. Percent contribution is how much training gain each variable provided to the model, and permutation importance is how much the training area under the curve (AUC) is impacted by a variable being permuted randomly. Area under the curve is a measure of how well a model can differentiate between presences and background points within the data and scales from 0 to 1, where values above 0.5 mean the model is better than random. Additionally, we calculated AUC on the test data used to evaluate model support.

## Fine-scale habitat modeling
### Field methods

In March 2019, we conducted a preliminary survey with hand excavation at 14 sites to ensure that *C. causeyi* was still present and detectable in its known range with the methods previously described in *Quebedeaux (2021)*. We received a scientific collection permit from AGFC (permit number 021520193) to sample in Arkansas, USA, and a permit waiver letter from The National Forest Service (file number 2600; 2700) to sample in the Ozark National Forest. Following this preliminary survey, we conducted standardized crayfish and habitat sampling over the summers of 2019 and 2020. We did not conduct conventional sampling during spring 2020, peak time for *C. causeyi* reproduction (*Robison, Crandall & Wagner, 2009*), due to COVID-19 pandemic travel restrictions. All historical sites that could be located (including known sites from March 2019 sampling) and new sites were sampled for crayfish and habitat. New sites were selected from across the entire anticipated range of *C. causeyi,* located near streams, and spanned a gradient of habitats predicted to range from unsuitable to highly suitable from our MaxEnt SDM model.
We sampled crayfish and gathered habitat data from 27 sites during May and June 2019 and 24 sites during June 2020 (51 total sites). We collected habitat data and crayfish from five, 1 m$^2$ quadrats placed at each site. We marked quadrats with a 1 m$^2$ polyvinyl chloride (PVC) pipe quadrat sampler and flags. We placed the first quadrat within the stream channel, or the dried stream channel, haphazardly. Then we placed a quadrat 10 m from the central quadrat upstream, downstream, on the left ascending bank, and on the right ascending bank. In situations where there was a barrier preventing a quadrat being placed exactly 10 m away from the central quadrat, we placed the quadrat as close to 10 m away as possible. Additionally, we conducted a 30 min timed search to aid in detection of *C. causeyi*, due to its low abundance at most sites. We searched anywhere in the vicinity of the quadrats, excluding the areas within the five quadrats. All crayfish were collected with hand excavation or dip net, which was used if there was standing water present and the burrows were found empty.

We characterized habitat within our 20 m × 20 m sites by recording the following variables: average forest canopy cover across quadrats, proportion of quadrats with large rock present, proportion of sand in soil at the central quadrat, presence of another primary burrowing species at the site (*Procambarus liberorum* Fitzpatrick, 1978), and the presence of stream dwelling crayfishes at the site (*Faxonius meeki* Faxon, 1898, and *Faxonius williamsi* Fitzpatrick, 1966). Low canopy cover was shown to be an important positive driver for occurrence of other burrowing crayfishes (*Rhoden, Taylor & Peterman, 2016*). We measured percent canopy cover at each quadrat using a Model-A spherical densiometer (Forestry Suppliers, Inc., Jackson, MS, USA), and averaged canopy cover from all five quadrats to represent canopy cover at the site scale. We expected the presence of large rock to have a positive effect on the presence of *C. causeyi* due to this species' association with large rocks in past surveys (*Robison, Crandall & Wagner, 2009*). We considered any rocks larger than 128 mm × 128 mm, including both cobble and boulders, to be large. Sandy soils have been shown to have a negative association with other burrowing crayfish species because they are not suitable for constructing burrows, and we expected to find the same with *C. causeyi* (*Dorn & Volin, 2009*; *Grow, 1982*; *Grow & Merchant, 1980*). Sand proportion was calculated from a soil core collected at the central quadrat at each site. We took only one soil core from the center of each site, anticipating little site-scale variability in soil attributes within this 20 m × 20 m area. The soil samples were analyzed with laser diffraction by the Illinois State Water Survey (Champaign-Urbana, IL, USA) (Table S1). We expected the presence of other crayfish species to have a negative effect on likelihood of *C. causeyi* occurrence due to competition (*James et al., 2015*; *Reynolds, Souty-Grosset & Richardson, 2013*). We separated crayfish species into other primary burrowing species or stream dwelling species as these organisms differ in their ecology and life history, and considered these crayfishes present or absent at the site if they were detected in any quadrat or by the timed search.

### Modeling

We modeled our fine-scale data with zero-inflated Poisson generalized linear models with the methods previously described in *Quebedeaux (2021)* in the program R (*v.* 3.5.2, *R Core*

*Team, 2018*) with the package glmmTMB (*v.* 1.0.2.1; *Brooks et al., 2017*). Our response variable was the number of quadrats within a site occupied by *C. causeyi*. We treated the timed search for *C. causeyi* as a sixth quadrat in this analysis because we were interested in modeling *C. causeyi* habitat at the site, rather than quadrat or micro-habitat, scale. We had intended to estimate *C. causeyi* occupancy while correcting for detection probabilities from our replicated quadrat samples (*Durso, Willson & Winne, 2011*; *MacKenzie et al., 2003*), but had too few detections for these models to converge. Accordingly, we instead used zero-inflated models to account for the large number of absences in our data. We used the habitat data collected from the quadrat sampling and the interpolated MaxEnt output to create a suite of potential models of *C. causeyi* relative abundance at a site scale. We ground-truthed the MaxEnt model by using the interpolated MaxEnt habitat suitability for the site as a predictor variable. Candidate models included the null model, sand only, interpolated MaxEnt habitat suitability, canopy cover only, proportion of quadrats containing a large rock, presence/absence of *P. liberorum*, presence/absence of stream dwelling crayfish species, sand and *P. liberorum*, interpolated MaxEnt output and sand and *P. liberorum*, and sand and large rock. We had no global model because the model would not converge. We selected the most supported model with Akaike's information criterion corrected for small sample size (AICc). Additionally, we analyzed the accuracy of our MaxEnt predictions by calculating the AUC of the receiver operating characteristic of *C. causeyi* occupied sites compared to unoccupied sites from our fine-scale habitat sampling (*Fawcett, 2006*; *Rhoden, Peterman & Taylor, 2017*) with the pROC package in program R (*Robin et al., 2011*).

## Environmental DNA
### Environmental DNA field collection
During March 16-18, 2020, we sampled 25 stream sites across northern Arkansas for *C. causeyi* eDNA. We intended on sampling all 51 sites where habitat data was collected, in addition to some of our preliminary sampling sites, but we were unable to do so as a consequence of COVID-19 pandemic restrictions on travel. Within sites sampled for eDNA, we included a combination of *C. causeyi* detections and non-detections from field sampling to evaluate assay performance for this species. We used one field blank of distilled water per site to assess for potential contamination in supplies. At every site, we triple-rinsed water bottles with stream water prior to water collection. We then collected four 250 mL surface water samples, spaced 10 m apart, along a 40 m reach of the stream by submerging the water bottle until full or by using the bottle cap to assist with collecting enough water in shallower streams. We targeted surface waters during the spring to detect *C. causeyi* in adjacent, submerged burrows or crayfish using surface water. We did not collect eDNA soil or water samples directly from crayfish burrows because the burrows were either in or next to the stream channel during the spring months in which we sampled, and *C. causeyi* are often found outside of burrows within the streams when they are reproductively active (*Robison, Crandall & Wagner, 2009*; *Robison & Leeds, 1996*). Additionally, studies have found that reproductive events may increase the eDNA concentrations in water samples, making target species easier to detect (*Curtis et al., 2021b*; *de Souza et al., 2016*; *Spear et al.,*

*2015*). Also, many of the ephemeral streams that *C. causeyi* utilize become dry during the summer. Ephemeral stream drying combined with the reproductive behavior of *C. causeyi* makes our eDNA sampling design specifically adapted for surface water sampling during spring months. After samples were collected, we sealed the bottles in a clean plastic bag and placed them on ice in a cooler until filtration (<12 h of sample collection) (*Curtis, Larson & Davis, 2021a*).

Prior to filtration of water samples, we cleaned the filtering space with 50% bleach solution. We then vacuum filtered water samples onto 1.0 μm cellulose nitrate filters (Whatman[TM]) and submerged the filters in ~1 mL of cetyl trimethylammonium bromide (CTAB; Teknova, Hollister, CA, USA) in a two mL centrifuge tube. We kept filters in the dark and at room temperature for about one month to increase eDNA concentrations by cell lysis (*Renshaw et al., 2015*), and then froze the filters at −80 °C until we extracted DNA. All bottles and filtration supplies had been washed with 50% bleach prior to use and we wore nitrile gloves to collect all water samples, filter samples, and during all subsequent lab procedures with frequent glove changes.

### *Cambarus causeyi* assay

We designed a single-species qPCR assay for *C. causeyi* by using the OligoAnalyzer[TM] Tool and the program *Sequencher* 5.4 (Gene Codes Corporation, Ann Arbor, Michigan, USA). We used *C. causeyi* sequences (accession numbers: JX514477.1 and DQ113443.1) from GenBank and newly sequenced individuals of the species (accession numbers: OP673577, OP673578, OP673579, OP673580, OP673581, OP673583, OP673584) to design the following primer-probe assay to amplify a 119 bp region of the cytochrome oxidase subunit one (COI) DNA bar coding region:

F-primer: 5′-GCGGGTATAACTATAGATC-3′
R-primer: 5′-CTATGCTATTAACAGATCG-3′
Probe: 5′-FAM-ACTATTGTTATCTTTGCCTGTG-MGB-NFQ-3′.

We tested this assay *in silico* against potential co-occurring non-target crayfish species (*Cambarus hubbsi* Creaser, 1931, *P. liberorum*, *F. williamsi*, *F. meeki meeki*, and *F. meeki brevis* Williams, 1952) with less than 95% similarity in sequence match (Table 1). Next, we tested the assay *in vitro* using synthetic COI gBlock fragments of *C. causeyi* and the four potential non-target species (Table S2).

We optimized the assay by testing different primer and probe concentrations and annealing temperatures until we produced the lowest Cq values, the highest efficiency and best $R^2$ using serial dilutions of the *C. causeyi* gBlock. We then tested the assay against high concentrations (0.57 −1.07 ng/μL confirmed concentration on Qubit[TM]) of non-target species DNA to examine if there was non-target amplification. *Faxonius williamsi* (0.79 ng/μL) and *P. liberorum* (0.57 ng/μL) DNA did not amplify, but there was delayed amplification in *F. meeki meeki* (0.69 ng/μL amplified in 3/3 plate replicates at Cq 27) and *C. hubbsi* (1.07 ng/μL amplified in 2/3 plate replicates at Cq 22). We then diluted the *F. meeki meeki* and *C. hubbsi* gBlock to a more environmentally realistic concentration ($10^{-4}$ ng/μL) for an eDNA sample and DNA samples did not amplify within the 40 qPCR cycles.

Quebedeaux et al. (2023), *PeerJ*, DOI 10.7717/peerj.14748

**Table 1  Comparison of primer-probe assay to non-target crayfish species.** Comparison of *Cambarus causeyi* primer-probe assay against non-target crayfish species. Locations of mismatches are underlined and in bold.

| Species | GenBank accession numbers | Forward primer mismatch | F primer percent identity | Reverse primer mismatch | R primer percent identity | Probe mismatch | Probe percent identity |
|---------|---------------------------|-------------------------|---------------------------|--------------------------|---------------------------|----------------|------------------------|
| *Cambarus causeyi* | JX514477.1 | GCGGGTATAACTATAGATC | 100% | CTATGCTATTAACAGATCG | 100% | ACTATTGTTATCTTTGCCTGTG | 100% |
| | DQ113443.1 | | | | | | |
| | OP673577 | | | | | | |
| | OP673578 | | | | | | |
| | OP673579 | | | | | | |
| | OP673580 | | | | | | |
| | OP673581 | | | | | | |
| | OP673583 | | | | | | |
| | OP673584 | | | | | | |
| *Cambarus hubbsi* | MG872957.1 | G **TA**GGTATAACTAT **G**GATC | 84% | CTATG **TT** G**TT** G**AC** G**GATCG | 79% | A **T**TATT **A**TTATCTTT **A**CCTGTG | 86% |
| *Procambarus liberorum* | KF827978.1 | G **TA**GG **G**ATAACTATAGATC | 84% | CTAT **A**CTATTAACAGA **C**CG | 89% | A **TT** G**TTGTTATCT **CT **A**CCTGTG | 82% |
| | KF827979.1 | | | | | | |
| *Faxonius williamsi* | OP673582 | GC **T**GG **G**ATAACTAT **G**GATC | 84% | CTAT **AT**TATTAAC **T**GATCG | 84% | A **TT** G**TT A**C**TATC A**TT **A**CCTGTG | 73% |
| *Faxonius williamsi* | AY701252.1 | GC **T**GG **A**ATAACTAT **G**GATC | 84% | CTAT **AT**TATTAAC **T**GATCG | 84% | A **TT** G**TT A**TTATCTTT **A**CCTGTG | 82% |
| *Faxonius meeki meeki* | AY701213.1 | G **TA**GGTATAACTAT **G**GATC | 84% | CTAT **AT**TATTAAC **T**GATCG | 84% | A **T**TATTGTTATCTTT **A**CCTGTG | 91% |
| *Faxonius meeki brevis* | AY701212.1 | G **TA**GG **A**ATAACTAT **G**GATC | 79% | CTAT **AT**TATTAAC **T**GATCG | 84% | A **T**TATT **A**TTATCTTTGCCTGTG | 91% |

### Environmental DNA extraction and qPCR

Prior to eDNA extraction or qPCR preparation, we cleaned the laboratory space with a 50% bleach solution and treated hoods with UV for 20 min. We extracted DNA from filters using a chloroform–isoamyl alcohol extraction procedure (*Renshaw et al., 2015*) in a clean room free from high-copy DNA and isolated from the PCR laboratory (*Goldberg et al., 2016*). At the end of the extraction procedure, we eluted eDNA samples with 100 µL of TE buffer (Invitrogen™) and froze at −80 °C until qPCR. One extraction blank was used for every ~12 eDNA samples to assess the level of lab carry-over contamination.

We ran 20 mL qPCR reactions with the following: 10 µL of iTaq Universal Probes Supermix (Bio-Rad, Hercules, CA, USA), 4.0 µL of molecular grade nuclease-free water (Invitrogen™), 1.0 µL (10 µM) of each primer, 1.0 µL (10 µM) of the probe, and 3 µL of eDNA. For the negative plate controls included on every plate, we replaced the 3 µL of eDNA with 3 µL of molecular grade nuclease-free water. We prepared all plates (qPCR strip tubes) in an isolated and spatially separate PCR product-free clean room. We then transported our samples to our PCR room where we added our *C. causeyi* gBlock standard using a serial dilution (1:10) from $10^{-4}$ to $10^{-10}$ ng/µL to create a standard curve for each plate. We next ran the plate on a QuantStudio 3 Real-Time PCR (Applied Biosystem®) with the following parameters: 95 °C for 5 min denaturation and 40 cycles of 95 °C for 30 s, 52 °C for 1 min, and 72 °C for 1 min. We ran all samples in triplicate and considered a true amplification if 1/3 plate replicates amplified before cycle 40. In order to confirm that there were no false positives, all eDNA samples that amplified were cleaned with ExoSap-It Express, Sanger sequenced at the University of Illinois' W.M. Keck Center, and then confirmed as *C. causeyi* using NCBI's BLAST (*Altschul et al., 1990*). To test whether samples displayed evidence of inhibition, we randomly selected an eDNA sample that did not amplify *C. causeyi* eDNA and where no crayfish were found (KBQ 20-77) (Table S3), spiked the sample with $10^{-5}$ ng/µL *C. causeyi* gBlock standard, and confirmed that there was no evidence of inhibition (<1 average ΔCq between the spiked sample and the standard).

# RESULTS

## Species distribution modeling

Our MaxEnt model's test AUC was 0.911, indicating good fit, with a standard deviation of 0.0342 (Fig. 1). Average annual precipitation was the best predictor of *C. causeyi* presence estimated from a jacknife analysis (Fig. 2). All other variables performed poorly. Percent contribution and permutation importance mirrored results of the jackknife analysis. Precipitation had a contribution of 65.0%, Euclidean distance from a stream had a contribution of 13.4%, elevation had a contribution of 12.0%, and all remaining variables had a contribution under 5%. We found a unimodal relationship between average annual precipitation and *C. causeyi* presence, which peaked at approximately 140 cm/yr (Fig. 3). *Cambarus causeyi* habitat suitability declined below this value, plateauing around 150 cm/yr.
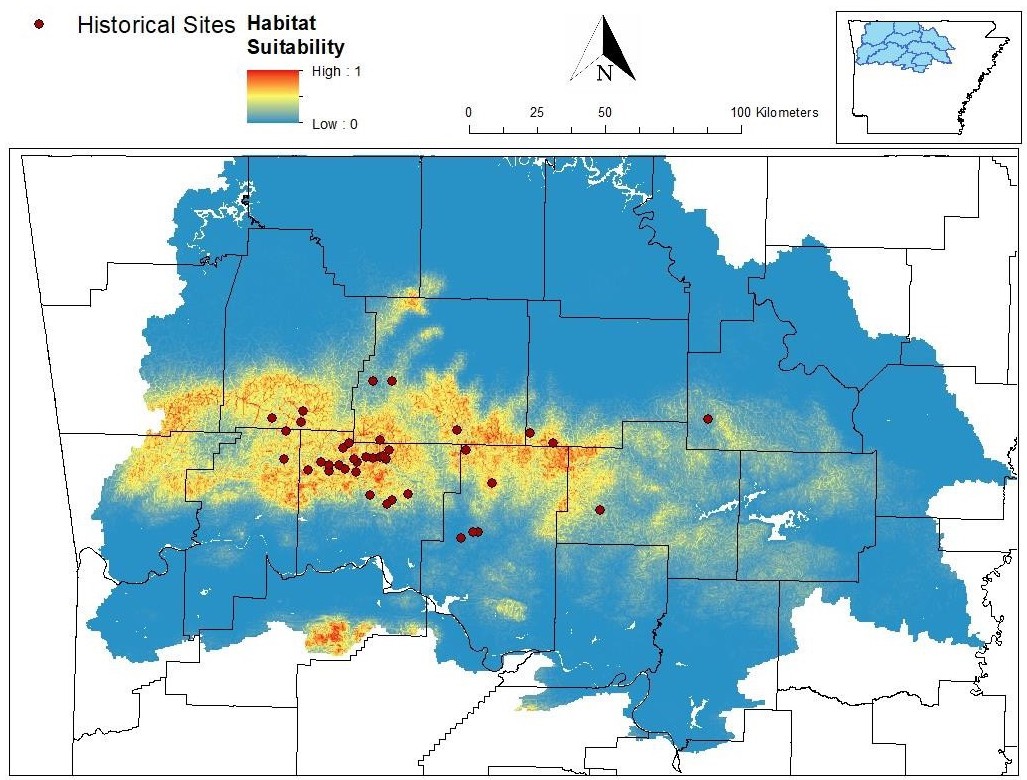

**Figure 1** **Predicted distribution of *Cambarus causeyi* from MaxEnt model.** Predicted distribution of *Cambarus causeyi* in Arkansas, USA from MaxEnt, all historical *C. causeyi* sites, and our modeling extent. HUC8 watersheds visualized in the inset map were used as the study extent because they overlapped with counties (black lines) with historical records.

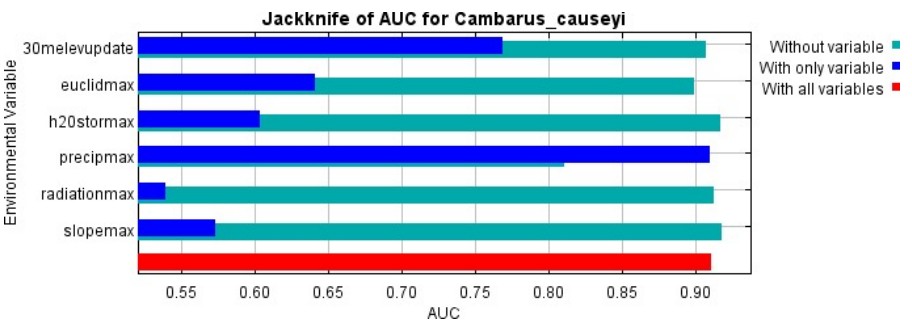

**Figure 2** **MaxEnt jackknife analysis results.** MaxEnt jackknife analysis on the area under the curve (AUC) for our *Cambarus causeyi* model. "30melevupdate" = elevation, "euclidmax" = Euclidean distance from a stream, "h2Ostormax" = average available water storage for the top 150 cm of soil, "precipmax" = 1981–2010 annual average precipitation, "radiationmax" = solar radiation, and "slopemax" = slope. Precipitation is the only variable that, when it was the only variable included in a model, had a higher AUC than in a model without that variable. Additionally, the AUC of the model with all variables was approximately equal to the AUC of the model with just precipitation.

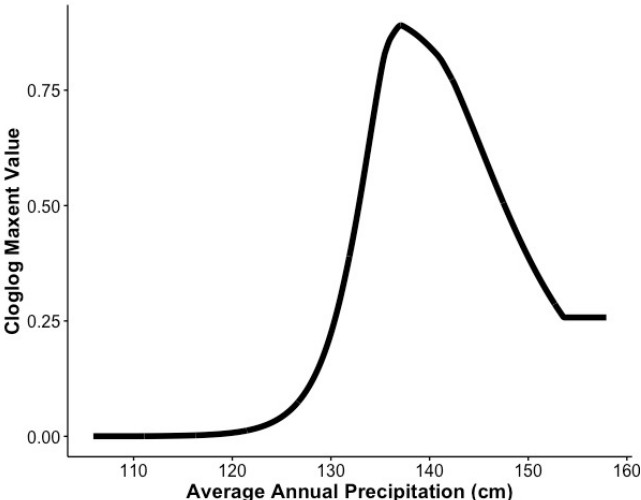

**Figure 3 Response of estimated suitability for *Cambarus causeyi* to precipitation.** The response of *Cambarus causeyi*'s estimated suitability, shown with the cloglog (complementary log–log) transformation, to the annual average precipitation in centimeters from 1981-2010 from the PRISM climate data (*Daly, Taylor & Gibson, 1997*) generated in our MaxEnt analysis.

## Fine-scale habitat modeling

We detected *C. causeyi* at nine of 51 sites during our habitat sampling (Fig. 4; Table S4). We failed to find *C. causeyi* at three sites where the species had been found during our March 2019 preliminary survey, and these preliminary detections were omitted from models (Table S3). We found male and female pairs sharing burrows during the spring preliminary sampling, and found one ovigerous female and many young instars during the summer habitat sampling. No models were particularly well-supported, and our most-supported model (proportion of sand and presence of *P. liberorum*) was only marginally better than the null model (Table 2). Both the proportion of sand and the presence of *P. liberorum* had negative associations with *C. causeyi* relative abundance (Fig. 5). Proportion of sand and presence of *P. liberorum* were also the only single variable models to perform better than the null model (Table 2). Interpolated MaxEnt habitat suitability generally performed poorly in predicting *C. causeyi* relative abundance from conventional sampling. Surprisingly, MaxEnt habitat suitability had a negative effect on *C. causeyi* relative abundance in all models including this variable (Table 2). Similarly, AUC for the field validation of our MaxEnt model was only 0.548, indicating that this model only performed marginally better than random.

## Environmental DNA

No field, extraction, or plate blanks amplified. Standard curves using *C. causeyi* serial dilutions had $R^2$ values between 0.980–0.998 and efficiency between 98–110%. Our limit of detection (LOD), or the concentration where 1/3 plate replicates amplified, was $1 \times 10^{-10}$ ng/µL (~0.77 copies). Our limit of quantification (LOQ), or the concentration where 3/3

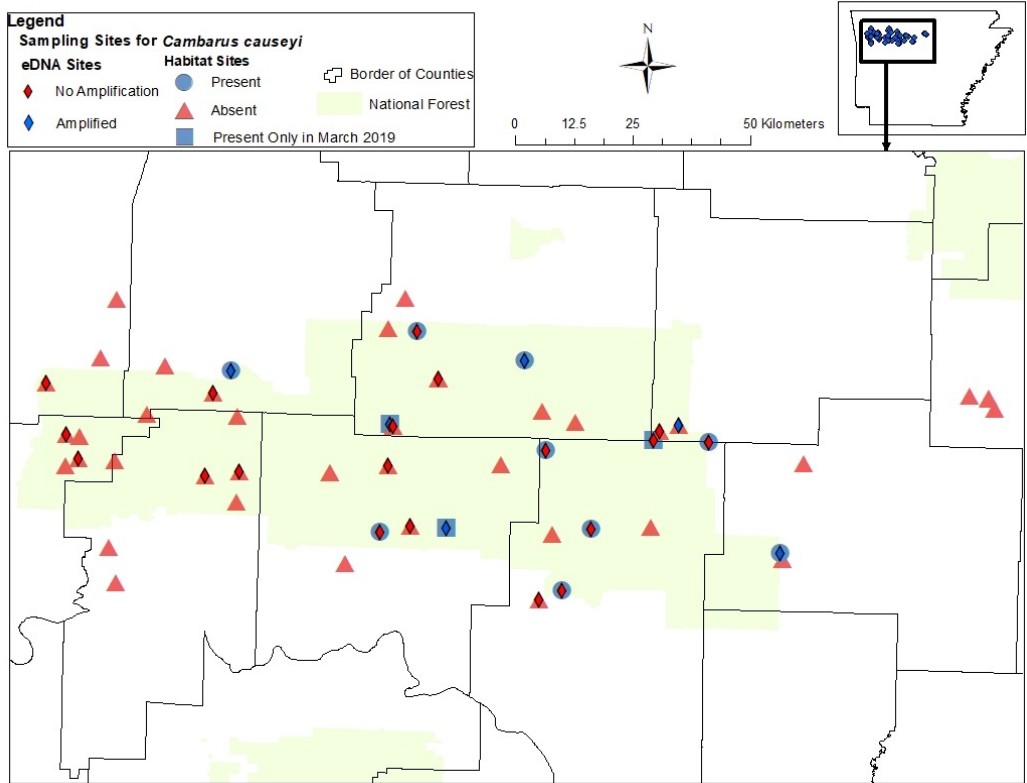

**Figure 4** **Map of sites sampled for fine-scale habitat modeling and eDNA.** All sites included in our fine-scale habitat modeling and eDNA sampling of *Cambarus causeyi* in Arkansas, USA. Red triangles were habitat sampling absences, and blue circles were presences during habitat sampling. Blue squares indicate sites where we detected *C. causeyi* in March 2019, but not during the course of our habitat sampling during the following summers. The squares were treated as absences in our fine-scale habitat analysis. Red diamonds are sites where *C. causeyi* was not detected with eDNA. Blue diamonds are sites where *C. causeyi* was detected with eDNA.

of the plate replicates amplified, was $1 \times 10^{-9}$ ng/µL ($\sim$7.73 copies). When detecting *C. causeyi* eDNA from field samples, concentrations were generally low (near LOD; Table S5).

We detected *C. causeyi* eDNA at six of the 25 stream sites sampled in March 2020 (Fig. 4; Table S3). These six detections included three sites where *C. causeyi* was collected during our fine-scale habitat survey, two sites where *C. causeyi* was collected in our March 2019 preliminary survey but not by fine-scale habitat sampling, and one site where *C. causeyi* was not collected by any conventional sampling. Non-detections of *C. causeyi* eDNA included six sites where the species was collected by our fine-scale habitat survey, and 13 sites where the species was not collected by any conventional sampling (Fig. 4). Accordingly, eDNA and conventional sampling for *C. causeyi* had moderate agreement, with a Cohen's Kappa statistic of 0.403 (five shared detections, one eDNA detection without conventional detection, six conventional detections without eDNA detection, and 13 shared non-detections). Environmental DNA detected *C. causeyi* more frequently (24%

**Table 2 Candidate models for our fine-scale habitat analysis.** Candidate models relating *Cambarus causeyi* relative abundance to predictor variables (Sand, *P. liberorum*, Large Rock, Interpolated Maxent, Stream Dwelling Crayfish, Canopy Cover) with the number of parameters (K), delta corrected Akaike Information Criteria (AICc), and log likelihood (LL) values representing model performance. Estimates of parameter coefficients with standard errors in parentheses are given for each predictor variable when it is included in a model, whereas predictor variables omitted from a model are represented by a dash (-).

| Sand | *P. liberorum* | Large Rock | Interpolated Maxent | Stream Dwelling Crayfish | Canopy Cover | K | ΔAICc | LL |
|---|---|---|---|---|---|---|---|---|
| −7.331 (5.001) | −20.401 (16810.675) | − | − | − | − | 4 | 0.00 | −27.107 |
| − | −20.860 (19040.000) | −2.196 (1.189) | − | − | − | 3 | 0.89 | −28.729 |
| −6.290 (3.634) | − | −2.084 (1.285) | − | − | − | 4 | 0.95 | −27.582 |
| −7.252 (4.977) | − | − | − | − | − | 4 | 1.02 | −27.616 |
| −7.832 (5.224) | −20.904 (19922.950) | − | −0.827 (1.430) | − | − | 3 | 1.71 | −29.144 |
| − | −20.443 (19897.237) | − | − | − | − | 5 | 2.12 | −26.938 |
| − | − | − | − | − | − | 2 | 2.40 | −30.618 |
| −7.728 (5.260) | − | − | −0.568 (1.468) | − | − | 3 | 2.83 | −29.701 |
| − | − | −1.631 (1.174) | − | − | − | 4 | 3.09 | −28.653 |
| − | −20.457 (19168.973) | − | −0.551(1.357) | − | − | 4 | 3.91 | −29.060 |
| − | − | − | − | −0.271 (0.916) | − | 3 | 4.57 | −30.574 |
| − | − | − | −0.244 (1.373) | − | − | 3 | 4.63 | −30.602 |
| − | − | − | − | − | −0.128 (1.158) | 3 | 4.65 | −30.612 |

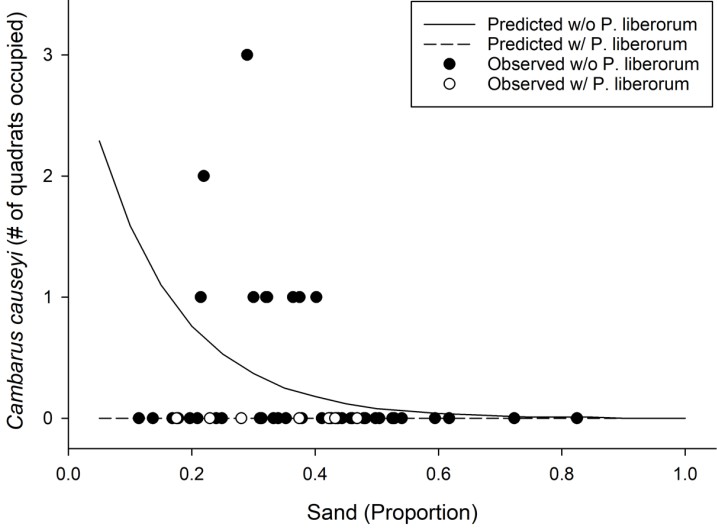

**Figure 5 Results from our most supported model in the fine-scale habitat analysis.** Observed (points) and predicted (lines) results from our most supported model of *Cambarus causeyi* relative abundance using *Procambarus liberorum* presence and proportion of sand as predictor variables (Table 2).

of sites) than conventional sampling overall (16.7% of sites excluding preliminary surveys, 17.7% including preliminary surveys).

## DISCUSSION

We assessed the distribution and habitat preferences of a rare, understudied crayfish species using SDMs, fine-scale modeling, and eDNA. Our MaxEnt model for this species appeared to perform well with a high test AUC and identified average annual precipitation as a factor in defining the distribution of *C. causeyi*. However, ground-truthing revealed poor performance of the MaxEnt model, as predicted habitat suitability was not included in any of our subsequent models explaining *C. causeyi* relative abundance. Instead, we found site-specific measures of soil texture and presence of another burrowing crayfish (*P. liberorum*) best explained *C. causeyi* relative abundance, albeit in relatively poor-performing GLM models. Large-scale analyses like MaxEnt and other SDMs often lack fine-resolution data and biotic interactions that can affect distributions (*Anderson, Peterson & Gómez-Laverde, 2002*; *Guisan & Thuiller, 2005*; *Peterman, Crawford & Kuhns, 2013*). Lastly, our eDNA assay detected *C. causeyi* from the field at a greater rate than, and in moderate agreement with, conventional sampling. Although SDMs and fine-scale habitat modeling from conventional surveys had moderate or weak performance explaining *C. causeyi* habitat associations, eDNA appears to be a promising approach to sampling for this and similar burrowing crayfish in the future.

We encountered several limitations during the course of our study. Our most supported generalized linear model from conventional field sampling performed better than the null model but had a mediocre fit (Fig. 5). We initially designed our field sampling data to be analyzed with occupancy models using replicated quadrats or timed searches for estimating detection probabilities (*Durso, Willson & Winne, 2011*; *MacKenzie et al., 2003*). However, we did not detect *C. causeyi* at enough sites for our models to converge. Instead, we employed zero-inflated Poisson generalized linear models on relative abundance of *C. causeyi*, but this approach was still limited by our infrequent crayfish detections. We had planned to improve detection of *C. causeyi* by sampling intensively in spring of 2020, but COVID-19 travel restrictions reduced our sampling in this season to only one week of eDNA surveys and no conventional surveys. We suggest that future sampling for *C. causeyi* be conducted in the spring when this species is most actively exhibiting mating behaviors (*Robison, Crandall & Wagner, 2009*), supported by our finding of male and female crayfish sharing burrows in March 2019 preliminary sampling but not in subsequent summer sampling. *Rhoden, Peterman & Taylor (2017)* had better success than our study in ground-truthing SDM models for burrowing crayfish species in Arkansas, but had sampled for these species in spring rather than summer. We expect this seasonal sampling recommendation also applies to eDNA sampling for *C. causeyi* and other burrowing crayfishes, as timing of eDNA sampling affects detection probabilities relative to life history and organism activity (*e.g.*, *de Souza et al., 2016*; *Troth et al., 2021*). For example, *Dunn et al. (2017)* found that ovigerous crayfish produce more eDNA than other individuals in a laboratory experiment. We suspect that *C. causeyi* would be most likely to be ovigerous in the middle to late spring, as we found many juveniles during our summer sampling but only one female with eggs.

Our MaxEnt model had a high testing AUC when validated on historic *C. causeyi* occurrences, but did not predict detections of this species well during ground-truthing in 2019 and 2020. MaxEnt identified average annual precipitation as the best predictor of *C. causeyi* presence, and this climate factor may limit the distribution of *C. causeyi* to wetter locations at a broad scale. However, our MaxEnt model seemingly missed factors affecting *C. causeyi* presence and relative abundance at the site or local scale. For example, in our fine-scale modeling, proportion of sand in soil was negatively associated with *C. causeyi* relative abundance, but water storage capacity of soil was not identified as an important variable in our MaxEnt model. Given these disparate results, we subsequently compared water storage capacity of soil from gSSURGO to our field-observed sand proportion at sites. Sandier soils should have low water holding capacity, but we found a very weak, noisy relationship between proportion of sand in soil and estimated water holding capacity from gSSURGO (Fig. S2). Soil conditions are important to burrowing crayfish (*Dorn & Volin, 2009*; *Grow, 1982*), but our MaxEnt model seemingly lacked high resolution or accurate soil data for our study sites. Improved spatial data on soil attributes (*i.e.*, *Vergopolan et al., 2022*) may lead to better SDM models for burrowing crayfish in the future.

Biotic interactions also appear to affect *C. causeyi* presence and relative abundance. *Cambarus causeyi* was never found at the same site as the primary burrowing crayfish *P. liberorum* (Fig. 5), which may be excluding *C. causeyi* by competition. *Egly & Larson (2018)* found poor ground-truthing performance of an SDM for crayfish to be partially explained by biotic interactions, in which native species were absent from suitable habitat occupied by invasive species. Biotic interactions can be incorporated into SDMs (*Anderson, Peterson & Gómez-Laverde, 2002*; *Guisan & Thuiller, 2005*; *Peterman, Crawford & Kuhns, 2013*), but we lacked a priori data to know that *P. liberorum* may be an effective competitor excluding *C. causeyi* from otherwise suitable sites. Our study emphasizes the importance of having familiarity with a study species' biology and ecological interactions when selecting variables for SDMs.

We designed and validated a robust qPCR assay for *C. causeyi* that does not amplify the DNA of non-target crayfishes at realistic, field concentrations. Five of the six eDNA detections for *C. causeyi* were at locations where the species was detected by conventional sampling, whereas only one eDNA detection was at a location where *C. causeyi* was not directly collected in our study. In this one instance, eDNA might be interpreted as out-performing conventional sampling for *C. causeyi*, particularly as we documented no evidence of field or laboratory contamination in our study commonly associated with false positives for eDNA studies. However, eDNA may have been advantaged relative to conventional sampling because eDNA samples were taken in the spring during the presumed active, reproductive season of *C. causeyi* (*Robison & Leeds, 1996*). Future comparisons of eDNA and conventional sampling for burrowing crayfish would ideally be done during the same season and year. Alternatively, we failed to detect *C. causeyi* eDNA from six sites where the species was detected by conventional sampling in either 2019 or 2020. These apparent non-detections or false negatives could be the consequence of mismatches between years of sampling eDNA and conventional surveys, especially because only one of the potential false negatives was conventionally sampled in 2020

(Table S3). For example, *C. causeyi* may have experienced population declines at some sites between conventional sampling in 2019 and eDNA sampling in 2020 (*Fournier, White & Heard, 2019*). Our *C. causeyi* eDNA samples had low copy numbers, near the LOD, when the species was detected, which may have been caused by low local abundance or biomass at most sites. Therefore, the performance of our eDNA assay for *C. causeyi* might also be improved by increasing either the volume of water filtered or number of field replicates taken (*Dougherty et al., 2016*; *Sepulveda et al., 2019*). Additionally, future burrowing crayfish monitoring could be done by testing burrow water or soil for eDNA (*Baker et al., 2020*), although sampling *C. causeyi* associated surface water (ephemeral streams) during the reproductive season appears to be a good approach for this particular species. Given the difficulty of detecting *C. causeyi* by conventional sampling, eDNA appears to be an encouraging approach for monitoring the presence of this species and other primary burrowing crayfish.

## CONCLUSIONS

Our study was motivated by the possibility that *C. causeyi* had experienced severe range and population declines between *Robison & Leeds (1996)* and *Robison, Crandall & Wagner (2009)*, which resulted in elevated IUCN conservation status for this species. Our results suggest that *C. causeyi* still occurs throughout its historic range (Fig. 4), but the species is difficult to detect. Further, detection probabilities for *C. causeyi* are likely higher in spring than in summer, and may be higher in wetter than drier years. For example, MaxEnt revealed a positive association between *C. causeyi* presence and wetter locations as represented by average annual precipitation. *Robison, Crandall & Wagner*'s (*2009*) *C. causeyi* survey was preceded by and coincided with the 2006–2007 extreme drought in Arkansas (*National Drought Mitigation Center, U.S. Department of Agriculture & National Oceanic and Atmospheric Administration, 2022*). Although *C. causeyi* is associated with higher elevation ephemeral and intermittent streams, which dry during a typical year, the species may still be negatively affected by severe drought. *Robison, Crandall & Wagner (2009)* may have detected widespread, drought-associated population declines of *C. causeyi* that the species has since recovered from. Alternatively, *C. causeyi* may also have been difficult to detect during *Robison, Crandall & Wagner (2009)* due to drought conditions. *Cambarus causeyi* uses shallow burrows, often under large rocks, but may have burrowed deeper to follow the receding water table during the 2006–2007 drought (*Novakowski & Gillham, 1988*; *Robison, Crandall & Wagner, 2009*). Future work could investigate seasonal and annual effects of precipitation on both *C. causeyi* occupancy and detection probability.

Although we did not find a decrease in the range of *C. causeyi*, there are still reasons this species may be vulnerable to population declines or range contraction. Future work might investigate biotic or competitive interactions with other burrowing crayfishes like *P. liberorum*, which could displace *C. causeyi* if they spread due to factors like climate or land use change. *Cambarus causeyi* could also be directly vulnerable to climate change if severe weather events like droughts become more common in Arkansas over time (*Mukherjee, Mishra & Trenberth, 2018*), particularly given the apparent preference of this species for

locations with high average annual precipitation. Despite our success finding *C. causeyi* at more sites than *Robison, Crandall & Wagner (2009)*, we suggest the species still deserves its vulnerable conservation status and increased management attention due to its narrow range, low local abundances (Table S3), and the threats described above. Furthermore, eDNA was effective at detecting *C. causeyi,* and future monitoring for this and other burrowing crayfishes might benefit from incorporating eDNA into sampling programs, although this emerging methodology should continue to be tested against conventional field sampling and distributional predictions from models like SDMs.

## ACKNOWLEDGEMENTS

We thank Brian Wagner, Caroline Caton, Caitlin Bloomer, Dustin Lynch, Justin Stroman, Matthew Anderson, Maxwell Hartman, and Katie Morris for assistance with field work. We thank Milton Tan for assistance with data analysis, and Mark Davis for use of eDNA lab space. This manuscript was improved by comments from Allyson Yarra and one anonymous reviewer.

### Funding

This work was supported by the USDA Hatch project ILLU-875-976 along with the Arkansas Game and Fish Commission State Wildlife Grant. Arkansas Game and Fish Commission employees volunteered time to help collect field samples.

### Grant Disclosures

The following grant information was disclosed by the authors:
USDA Hatch project ILLU-875-976 along with the Arkansas Game and Fish Commission State Wildlife Grant.

### Competing Interests

The authors declare there are no competing interests.

### Author Contributions

- Kathleen B. Quebedeaux conceived and designed the experiments, performed the experiments, analyzed the data, prepared figures and/or tables, authored or reviewed drafts of the article, and approved the final draft.
- Christopher A. Taylor conceived and designed the experiments, performed the experiments, analyzed the data, authored or reviewed drafts of the article, and approved the final draft.
- Amanda N. Curtis conceived and designed the experiments, performed the experiments, analyzed the data, prepared figures and/or tables, authored or reviewed drafts of the article, and approved the final draft.
- Eric R. Larson conceived and designed the experiments, performed the experiments, analyzed the data, prepared figures and/or tables, authored or reviewed drafts of the article, and approved the final draft.

### Field Study Permissions

The following information was supplied relating to field study approvals (*i.e.*, approving body and any reference numbers):

We received a scientific collection permit from Arkansas Game & Fish Commission (permit number 021520193), and we received permission from the US Forest Service to sample in the Ozark National Forest (file number 2600; 2700).

### DNA Deposition

The following information was supplied regarding the deposition of DNA sequences:

Sequences are available at GenBank: OP673577, OP673578, OP673579, OP673580, OP673581, OP673582 OP673583, OP673584.

### Data Availability

The code from our fine-scale habitat modeling and raw data (including raw soil data, habitat data, sampling sites, and data on our eDNA amplifications) are available in the Supplemental Files.

### Supplemental Information

Supplemental information for this article can be found online at http://dx.doi.org/10.7717/peerj.14748#supplemental-information.

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
