# Peer review of "A multi-method approach for assessing the distribution of a rare, burrowing North American crayfish species"

_PeerJ, doi:10.7717/peerj.14748_

## Round 0.1 · original submission · Minor Revisions

This paper has been reviewed by 2 reviewers, and they have included helpful comments that I think will improve the paper. In particular, please pay attention to the comments from Reviewer 2 about the habitat model / quadrats.

·

Basic reporting

This article was written in a clear and concise manner with professional English used throughout. I noted few minor grammatical errors (see below). The structure of the article conforms to an acceptable format with all necessary sections included. All figures, tables, and supplemental data are relevant and accessible. All field study permits are mentioned and appear correct. Results directly coincide with the objectives of the study described in the introduction.

The authors demonstrated a deep understanding of previous studies surrounding crayfish ecology and eDNA methods and cite these properly. The relevance of this study is clearly demonstrated by the well-formulated introduction and it is very clear how this work fits into the broader field of knowledge. I recommend placing a greater emphasis on the vulnerability of freshwater crayfishes, particularly, primary burrowers. This is alluded to through mentions of C. causeyi’s endemism, but the concept may be more solidified with a few sentences to explain how dire the situation is for freshwater crayfishes in general. One article that illuminates this is Wilcove and Master 2005. Based on available knowledge, about half of all assessed species of freshwater crayfish are considered imperiled, making crayfish the second most imperiled group of freshwater invertebrates (after unionid mussels and gastropod snails). This addition to the introduction may help to strengthen the argument as to why studying and attempting to conserve endemic primary burrowing crayfish is of extreme importance.

Line specific suggested changes:
Ln 99: change to “[an] understudied and rarely observed burrowing crayfish…”
Lns 101-104: consider splitting this sentence into two sentences for clarity

Experimental design

To the authors’ knowledge, this study is the first to implement a successful eDNA assay for a terrestrial burrowing crayfish. This, coupled with the fact that primary burrowing crayfish are critically understudied and imperiled adds to the study’s novelty and relevance. The authors clearly identify a gap in our understanding of narrow-ranged endemic crayfish, particularly primary burrowers, in the Ozark Highlands, and apply an established technique (eDNA assay) to aid in its conservation. Moreover, information regarding C. causeyi ecology and habitat preferences/requirements was explored, which adds to the limited understanding of Ozark crayfishes.

All methods were described with sufficient detail and could certainly be reproducible by others. The justification for the use of the statistical tests chosen was extremely well-reasoned and all methods chosen were appropriate to address the objectives at hand. One element of the analysis that should not be overlooked was the use of ground-truthing the SDM predictions with conventional sampling. Through the use of in situ presence and habitat data, it was determined that, despite the MaxEnt model performing well and suggesting a relationship between precipitation and C. causeyi distribution, this was not what the authors observed from field data. The choice to validate the SDM models with empirical data was extremely valuable and illustrates the need for the field to encourage the continuation of hands-on data collection, taxonomy, and natural history, all of which can provide information that cannot be gleaned from statistical modeling alone.

No suggested improvements

Validity of the findings

The authors directly acknowledged some of the limitations present in the study. For example, the pandemic affected their ability to collect habitat data at all sites. In terms of statistical analyses, their small sample size and limited number of detections was taken into consideration appropriately (e.g., through the use of zero-inflated models, and the use of MaxEnt which has been shown to be reliable with small sample sizes). Moreover, the authors recommend future research, particularly in the context of biotic interactions between C. causeyi and sympatric species as the present study did not include these considerations. Appropriate conclusions relevant to the study objectives were drawn from the results and there were no claims of causative relationships. Datasets were provided and appear complete.

There were a few points made in the discussion section of the article that I would have liked some clarification on, largely related to eDNA sample collection and its implications. Please note that the following comments are largely due to the fact that I am not experienced with eDNA methods myself, and I imagine other aquatic ecologists who may also not have this background would benefit from additional clarification.

Could you elaborate on why you sampled water from the stream and not from water and/or soil in the burrows? You mentioned in lns 276-277 that eDNA was not collected from soil or water directly from burrows since that approach may be more vulnerable to PCR inhibition and cited Baker et al. 2020. Baker et al. collected water samples directly from the burrows, but did not collect soil samples. Might more eDNA be present in an area less dynamic than a stream? Could soil potentially retain a signature longer or more effectively than flowing water? Since C. causeyi rarely leaves its burrow, wouldn’t you expect more genetic material to accumulate there? I am generally just curious as to why the stream was chosen to sample. The burrows extend down into the water table, but would this be sufficient enough to detect from a surface water sample? Perhaps soil just isn’t suitable, but it would be helpful to elaborate a bit more on this either in the methods or discussion.

In the discussion, you mention that life history likely plays a role in the effectiveness of eDNA sampling, and that future studies should attempt sampling in the spring when C. causeyi is most active due to mating. Why would crayfish activity affect eDNA detection? Wouldn't you assume that individuals are in the vicinity and would be detected regardless of sharing a burrow with a mate, etc? Or, is it possible that more “shedding” occurs the more active an organism is, or perhaps you are hoping to capture eDNA following crayfish movement from one burrow to another. A few sentences of clarification would be helpful.

Additional comments

I commend the authors for applying a new technique to aid in the detection and conservation of a cryptic, endemic Ozark crayfish and for producing a statistically-sound and thorough study. I recommend this article for publication in Peer J - Life and Environment following the completion of minor revisions I suggest in my review.

Reviewer 2 ·

Basic reporting

The manuscript flowed nicely from introduction to conclusion. However, the writing is wordy. I have given examples in line-by-line texts of ways to reduce wordiness. Literature references with sufficient field background/context were provided. Raw data was provided and relevant results were provided from hypotheses.

Experimental design

This paper meets the aims and scopes of PeerJ, with focus on biological science. The research question is well defined, relevant and meaningful and they state how research fills the identified knowledge gap. Enough information was given to be a repeatable study.

This study evaluates 3 different approaches in assessing the distribution of a rare primary burrowing crayfish. This manuscript increases the knowledge of the suitability of using a common modeling approach, conventional sampling/habitat models, and eDNA on primary burrowing crayfishes. Because primary burrowing crayfishes are hard to sample it is beneficial to have alternative tools to assess their habitat use and range, so assessing the use of maxent models and eDNA for assessing primary burrowing crayfish distributions will help guide management and others on suitable tools to use when assessing this species as well as other similar primary burrowing species.

I question the validity of the habitat model. This model uses burrow counts from outside of the quadrats sampled, but there was not habitat data taken outside of those quadrats. I’m unsure how the habitat model was run without this data for all “quadrats”. Either something is missing in the text that describes habitat data taken outside of the quadrats or this model was run incorrectly, and only burrow counts inside quadrats should be used.

Validity of the findings

Overall, this is a good study that will begin to fill a knowledge gap and be directly useful to management. Findings from this study will help guide management of when and when not to use maxent models, the types of data needed to make useful maxent models and the reliability of this tool. Because burrowing crayfish are hard to collect, the use of eDNA is also another tool that would be beneficial for understanding burrowing crayfishes distributions, and this study provides insight on the time of year this would be useful and the differences between findings from eDNA and traditional sampling. This study as shows the type of steps required to develop a useful eDNA marker. This study increases the knowledge and understanding of the distribution of a rare primary burrowing crayfish.

All underlying data has been provided; they are robust, statistically sound, & controlled. However, some conclusions are overstated and should be scaled down to clearly link to supporting results. Other statements should be made within the discussion stating the short comings of the study and things that could have impacted the results.

Additional comments

TITLE
Because this study focuses on crayfish distribution and does not explicitly say what measures should be taken to conserve the species, I suggest changing the title. A more appropriate title may be: “A multi-method approach for assessing the distribution of a rare, primary burrowing North American crayfish species”

METHODS
A number of hypotheses are stated throughout the methods section which breaks up the flow of what steps was taken in each of your analyses. It will be easier to follow the methods section if you take out your hypotheses and place them in a separate paragraph. You should also use your hypotheses in the discussion section to discuss the differences between what you hypothesized and what was found.

Line 176— In Fig. S1, please highlight the watersheds used in the MaxENT model. From your figure it looks as if 4 watersheds would be omitted but in the texts you state only 1 was omitted. Please explain show in the figure which was omitted and explain why only 1 was omitted instead of the 4 that don’t include historical samples.
Line 178-191 I appreciate the thorough explanation of the maxent model.
Line 201-203: This line can be shorted to simple say “We did not take habitat samples during the spring, C. causeyi peak time for activity (Robison et al. 2009), due to COVID-19 pandemic travel restrictions.
Would not sampling during their peak time decrease your ability to fine them when ground-truthing?
226-227: This should be in your discussion not methods section.
228-230: Move to discussion section or a hypotheses paragraph.
231: add “large cobble” behind your 128 mm x 128 mm.
231-234: Move to discussion section or a hypotheses paragraph.
236-238: Move to discussion section or hypotheses paragraph.
Information from 152-176 that state your hypotheses can also be added to this hypotheses paragraph or in the discussion section.
243-244: Did you have habitat data for areas outside of the quadrat? If not how can it be added to the model?
244-248: This can simply be stated as “Because we had too few detections, we used zero inflation models to account for the large number of absences in our data.”
251-255: This information should be in your results section. It may be best to put it in a table and show the parameters and AIC values for each model.
266-268: This can simply state “ A subset of sites were sampled due to COVID-19 pandemic travel restrictions.”
268: Reduce wording to “Within sampled sites,…..”
274-276: Remove hopefully from sentence “We targeted surface waters during the spring to increase our likelihood of detecting C. causeyi in adjacent, submerged burrows or juvenile crayfish using surface water.
315-316: If F. meeki and C. hubbsi were in high concentrations could you get a false positive? If so this should be discussed in your discussion. If not, it would be helpful to say over what concentration of F. meeki and C. hubbsi iwould be needed to get a false positive.
337: 95C (missing degrees)

RESULTS
351-359: FigS2 should be a main figure in the manuscript instead of supplemental. Please add the curves associated with the models to this figure.
363: Figure 3 its hard to see the stars in the triangles, please make edits to size, colors, or shapes.
366: Change “but” to “and”
371-373: This data can be added to table 2.
383: Another sentence that gives an over of what this paragraph will discuss should start this paragraph,

DISCUSSION
409-411: You stated in your results that this model was almost the same as the random model so none of your variables were highly correlated with C. causeyi abundance. So I think habitat wise it still needs to be figured out what’s driving C. causeyi distributions. Please change your statement to reflect this.
415-417: I think this statement is overreaching. Your models were not useful in predicting C. causeyi presence or abundance. I do think from this you show more habitat analyses are needed but eDNA may be a way to sample sites rather than conventional sampling methods.
343-436: You should clearly state that there is a chance that you found differences between eDNA and conventional sampling detection due to timing of sampling. Taking eDNA in the spring should have increased detection possibilities and conventional sampling in summer likely decreased detection possibilities.
413-414: Did eDNA do better because you took eDNA samples in March but conventional samples in summer?
445-449: This figure should be moved to supplemental data.
469-476: Please add size and numbers of crayfish collected to your results. Were sites where you didn’t get eDNA detections at sites where there was a lower abundance of crayfishes or smaller crayfishes collected during conventional methods?
492-495: This should be discussed in line 438-463 as to the differences between the environment in historical and current studies which may have driven the importance of precipitation in the model.

Figures
Figure 2: Cloglog is not mentioned in the text. Please explain what it is.
Figure 5: R2 should be R2. Use 2 significant digits for your p value.

---

## Round 0.2 · accepted · Accept

Thanks for making the requested edits, and congratulations!